# Host Cells Upregulate Phosphate Transporter PIT1 to Inhibit *Ehrlichia chaffeensis* Intracellular Growth

**DOI:** 10.3390/ijms25147895

**Published:** 2024-07-19

**Authors:** Meifang Li, Nan Yang, Xiaoxiao Li, Nan Duan, Shanhua Qin, Mengyao Wang, Yuhong Zhou, Yongxin Jin, Weihui Wu, Shouguang Jin, Zhihui Cheng

**Affiliations:** 1The Key Laboratory of Molecular Microbiology and Technology, Ministry of Education, Nankai University, Tianjin 300071, China; 2120211156@mail.nankai.edu.cn (M.L.); 2120211170@mail.nankai.edu.cn (N.Y.); 15230183753@126.com (X.L.); duannan@mail.nankai.edu.cn (N.D.); qinshanhua2000@163.com (S.Q.); 2120231469@mail.nankai.edu.cn (M.W.); 2120231471@mail.nankai.edu.cn (Y.Z.); yxjin@nankai.edu.cn (Y.J.); wuweihui@nankai.edu.cn (W.W.); nksjin@nankai.edu.cn (S.J.); 2Department of Microbiology, College of Life Sciences, Nankai University, Tianjin 300071, China

**Keywords:** *Ehrlichia chaffeensis*, PIT1, Ech_1067, NF-κB, MyD88, innate immune responses

## Abstract

*Ehrlichia chaffeensis* infects and proliferates inside monocytes or macrophages and causes human monocytic ehrlichiosis (HME), an emerging life-threatening tick-borne zoonosis. After internalization, *E. chaffeensis* resides in specialized membrane-bound inclusions, *E. chaffeensis*-containing vesicles (ECVs), to evade from host cell innate immune responses and obtain nutrients. However, mechanisms exploited by host cells to inhibit *E. chaffeensis* growth in ECVs are still largely unknown. Here we demonstrate that host cells recognize *E. chaffeensis* Ech_1067, a penicillin-binding protein, and then upregulate the expression of PIT1, which is a phosphate transporter and transports phosphate from ECVs to the cytosol to inhibit bacterial growth. We found that host cells upregulate the PIT1 expression upon *E. chaffeensis* infection using transcriptome sequencing, qRT-PCR and Western blotting, and PIT1 is localized on the ECV membrane in infected THP-1 cells using confocal microscopy. Silence of PIT1 using shRNA enhances *E. chaffeensis* intracellular growth. Finally, we found that *E. chaffeensis* Ech_1067 induces the upregulation of PIT1 expression through the MyD88-NF-κB pathway using recombinant protein for stimulation and siRNA for silence. Our findings deepen the understanding of the innate immune responses of host cells to inhibit bacterial intracellular growth and facilitate the development of new therapeutics for HME.

## 1. Introduction

*Ehrlichia chaffeensis* is an obligatory intracellular bacterium that proliferates inside monocytes or macrophages and causes human monocytic ehrlichiosis (HME) [1]. *E. chaffeensis* is transmitted by tick and its major reservoir is white-tailed deer [2,3,4]. HME is generally characterized by anorexia, chills, headache, myalgia, and acute fever accompanied by the elevation of serum hepatic aminotransferases [5]. The severity of HME can range from being asymptomatic to fatal [5]. HME is particularly threatening to individuals who are immunocompromised or elderly [5,6]. HME can be treated with the broad-spectrum antibiotic doxycycline, which has significant side effects, and there are no vaccines available for this disease [1]. 

*E. chaffeensis* infects and multiplies in monocytes or macrophages, which are primary immune cells [7]. After internalization, *E. chaffeensis* resides in specialized membrane-bound inclusions, *E. chaffeensis*-containing vesicles (ECVs), which have early endosome-like characteristics [8,9]. ECV is segregated from the innate immune system and has an extra layer to protect bacteria from antimicrobial drugs [10]. *E. chaffeensis* has evolve specific mechanisms to manipulate intracellular vesicular trafficking to inhibit ECV maturation and its fusion with lysosome as well as promote autophagy and its fusion with autophagosomes [11,12]; thus, *E. chaffeensis* evades from host cell innate immune responses and obtains nutrients. However, mechanisms exploited by host cells to inhibit *E. chaffeensis* growth in ECVs are still largely unknown.

Phosphate is indispensable for living organisms and plays a crucial role in the metabolism of carbohydrates, proteins, nucleic acids and membrane lipids [13,14]. In mammals, five phosphate transporters on the cell membrane transport phosphate from environment to the cytosol [15]. Among these transporters, PIT1 is located on the membrane of bacteria-containing vesicles after infection and transports phosphate from vesicles to the cytosol as the extracellular side of the cell membrane faces the lumen of vesicles [16,17]. As an immune response, host cells upregulate PIT1 expression to reduce phosphate concentration in bacteria-containing vesicles and inhibit bacterial growth [16,17]. However, whether PIT1 is localized on the ECV membrane or plays a role in the inhibition of *E. chaffeensis* growth remains to be illustrated. In this study, to gain insights into PIT1 function, we performed transcriptome sequencing and Western blotting to examine its expression upon infection, immunofluorescent assay to identify its distribution and silenced PIT1 with shRNA to determine its role in *E. chaffeensis* intracellular growth. Then, we investigated the signal and pathway by which host cells induce the upregulation of PIT1 expression to inhibit *E. chaffeensis* intracellular growth. 

## 2. Results

### 2.1. Host Cells Upregulate the PIT1 Expression upon E. chaffeensis Infection

After infection of *E. chaffeensis*, host cells change the expression of various genes [18]. To obtain a whole picture of gene expression changes in host cells, we performed transcriptome sequencing of uninfected and *E. chaffeensis*-infected THP-1 cells. The results of transcriptome sequencing showed that the expression of total 2988 genes was changed, among which 1585 genes were upregulated and 1403 genes were downregulated in *E. chaffeensis*-infected THP-1 cells (Appendix A). EuKaryotic Orthologous Groups (KOG) analysis revealed that these genes could be grouped into signal transduction mechanism, transcription, post-translational modification and various other functions (Figure 1).

Notably, among the upregulated genes, *PIT1* was upregulated up to 3.02-fold. To verify this result, we performed qRT-PCR and Western blotting. At 48 h post infection (p.i.), we found the mRNA level and protein level were significantly higher in *E. chaffeensis*-infected cells compared to uninfected cells (Figure 2a,b). These results indicate that *E. chaffeensis* infection induces the upregulation of PIT1 expression in THP-1 cells.

### 2.2. PIT1 Is Localized on the ECV Membrane in Infected THP-1 Cells

*E. chaffeensis* enters THP-1 cells via receptor-mediated endocytosis [19,20]. ECVs are formed by engulfing *E. chaffeensis* surrounded by the cell membrane, it is likely that PIT1 is also located on the ECV membrane. We then investigated the distribution of PIT1 in host cells after *E. chaffeensis* infection. Confocal immunofluorescence microscopy results demonstrated that PIT1 was colocalized with *E. chaffeensis* in infected THP-1 cells (Figure 3), indicating that PIT1 is localized on the ECV membrane.

### 2.3. PIT1 Inhibits E. chaffeensis Intracellular Growth in Host Cells

PIT1 on the cell membrane transports phosphate from the extracellular side to the cytosol side. As for ECVs, the extracellular side of the cell membrane faces the lumen of vesicles. The upregulation of PIT1 expression may reduce phosphate concentration in ECVs and inhibits *E. chaffeensis* intracellular growth. We then investigated whether host cell PIT1 affects *E. chaffeensis* intracellular growth. We silenced PIT1 in THP-1 cells using shRNA (Figure 4a), and then infected with *E. chaffeensis.* We found that silence of PIT1 had no effect on the bacterial number at 4 h p.i. (Figure 4b), indicating PIT1 is not involved in the adherence and internalization of *E. chaffeensis*. At 48 h p.i., the bacterial number in PIT1-silenced cells was significantly higher compared to that in control cells (Figure 4b), indicating that PIT1 inhibits *E. chaffeensis* intracellular growth. 

### 2.4. Host Cells Upregulate the PIT1 Expression in an NF-κB-Dependent Manner

In bone marrow-derived macrophages (BMDMs), *PIT1* expression is regulated by the NF-κB pathway [21]. We then investigated whether the upregulation of *PIT1* expression in THP-1 cells upon *E. chaffeensis* infection is also regulated in an NF-κB-dependent manner. We silenced p65 in THP-1 cells using siRNA (Figure 5a), and then infected these cells with isolated *E. chaffeensis*. We found that the upregulation of *PIT1* expression was observed in control THP-1 cells but was significantly inhibited in p65-silenced THP-1 cells (Figure 5b). These data indicate that the upregulation of *PIT1* expression upon *E. chaffeensis* infection is regulated in an NF-κB-dependent manner.

### 2.5. Host Cells Recognize E. chaffeensis Ech_1067 to Upregulate the PIT1 Expression

Although *E. chaffeensis* lacks all of the genes for lipopolysaccharide biosynthesis and most of the genes for peptidoglycan biosynthesis [22,23], host cells recognize *E. chaffeensis* Ech_1067, a penicillin-binding protein, and upregulate the expression of cytokines in a MyD88-dependent manner [24]. MyD88 can activate NF-κB [25]. Then, we hypothesized that the upregulation of *PIT1* expression in THP-1 cells may be induced by *E. chaffeensis* Ech_1067. We expressed and purified recombinant Ech_1067 with an N-terminal His-tag (Appendix A), as well as recombinant *E. chaffeensis* Tr1, a regulatory protein that is not on the bacterial membrane [26], as a negative control (Appendix A). We first stimulated THP-1 cells with rEch_1067 to determine its ability to induce the *PIT1* expression in THP-1 cells. To eliminate stimulation from the endotoxin, the purified recombinant proteins were cleaned with the endotoxin removal kit and THP-1 cells were treated with polymyxin B sulfate (PB) [24]. The results showed that the *PIT1* expression in THP-1 cells was induced by rEch_1067 in a dose-dependent manner, while rTr1 had no effect (Figure 6). 

We then investigated whether MyD88 plays a role in the upregulation of *PIT1* expression. We silenced MyD88 in THP-1 cells using siRNA (Figure 7a) and then infected these cells with isolated *E. chaffeensis*. We found that the upregulation of *PIT1* expression was observed in control THP-1 cells, but was significantly inhibited in MyD88-silenced THP-1 cells (Appendix A). Then, we stimulated p65- or MyD88-silenced cells with rEch_1067 and found that the upregulation of *PIT1* expression was observed in control THP-1 cells, but was significantly inhibited in p65- or MyD88-silenced THP-1 cells (Figure 7b,c). These data indicate the upregulation of *PIT1* expression induced by rEch_1067 is through the MyD88-NF-κB pathway. 

## 3. Discussion

In this study, we found that after infection, THP-1 cells upregulate PIT1 expression, and PIT1 is located on the ECV membrane to transport phosphate from ECVs to the cytosol, which inhibits *E. chaffeensis* growth. Phosphate enters cells via five phosphate transporters [15]. PIT1 and PIT2 are expressed ubiquitously in all tissues, and the other three transporters are present in the kidneys and the gut [27]. In different types of cells, the expression of PIT1 is significantly higher than that of PIT2 [16,17]. After internalization, uropathogenic *Escherichia coli* (UPEC) resides in vesicles of bladder epithelial cells [17]. These cells upregulate the PIT1 expression, which is located on the membrane of UPEC-containing vesicles, to transport phosphate inside these vesicles to the cytosol [17]. *Salmonella enterica* serovar Typhimurium infects HeLa cells and induces the upregulation of PIT1 expression, which also reduces the phosphate concentration in *Salmonella*-containing vacuoles [16]. Considering the importance of phosphate during bacterial growth, it is highly likely that upregulation of PIT1 expression is a common response of host cells to combat bacterial infection, which might provide specific targets to develop new broad-spectrum therapeutics for bacterial infection.

Bladder epithelial cells or HeLa cells sense LPS of UPEC or *S.* Typhimurium and upregulate the PIT1 expression through TLR4-MyD88-NF-κB [16,17]. We found that the upregulation of PIT1 expression is through the MyD88-NF-κB pathway. The genome size of *E. chaffeensis* is only 1.18 Mb, and *E. chaffeensis* lacks flagella, pili, LPS, peptidoglycan [22,23]. The absence helps *E. chaffeensis* to escape from the recognition of Toll-like receptors (TLRs) and nucleotide-binding oligomerization domain-containing intracellular protein receptors (NLRs), which induce profound innate immune responses [1]. The proteins located on *E. chaffeensis* membrane provide a critical interface between the bacterium and its host cells [19,28,29,30,31,32]. We found that host cells recognize Ech_1067 and elicit immune responses. Thus, penicillin-binding proteins might be bacterial components recognized by host cells for innate immune responses. The receptor recognizing Ech_1067 remains to be investigated. Since *E. chaffeensis* is mostly dominant in immunocompromised individuals [5,6], in vivo experiments using immunocompromised or MyD88^−/−^ animals will provide more comprehensive information for the mechanism demonstrated here.

*E. chaffeensis* Ech_1067 is highly expressed and localized on the bacterial membrane [24], as the majority of other genes involved in peptidoglycan synthesis are missing [22], the function of Ech_1067 is unlikely in peptidoglycan biosynthesis. Ech_1067 is upregulated before the onset of exponential growth stage and highly expressed throughout the growth [24], suggesting an important role of Ech_1067 for the establishment of bacterial infection and growth, even Ech_1067 can be recognized by host cells. Ech_1067 induces interleukin 8 or CXCL2, tumor necrosis factor α, interleukin 1β, and interleukin 10 in THP-1 cells and mouse bone marrow-derived macrophages [24]. The induction of cytokines might be a key element for *E. chaffeensis* survival and persistence.

Bacteria have evolved sophisticated mechanisms to counteract the reduction of phosphate in bacteria-containing vesicles resulting from the upregulation of PIT1 expression. UPEC senses the reduction of phosphate via a two-component system, PhoB/PhoR, and activates the expression of *pldA* encoding a phospholipase, which disrupts the vesicle membrane and helps UPEC to escape into the cytosol [17]. *S*. Typhimurium also senses the reduction of phosphate by PhoB/PhoR and activates the expression of SPI-2 genes to promote *S.* Typhimurium’s virulence and contribute to avoiding the fusion of *Salmonella*-containing vacuoles with lysosomes [16,33]. The effectors of *E. chaffeensis* type IV secretion system (T4SS) induce autophagy and promote the fusion of ECVs with early autophagosome to obtain host cell-derived nutrients [11,12], which might be the mechanisms exploited by *E. chaffeensis* to acquire enough phosphate for its growth. However, how *E. chaffeensis* senses phosphate reduction is still unclear. *E. chaffeensis* has only three pairs of two-component systems, CckA/CtrA, NtrY/NtrX, PleC/PleD, but not PhoB/PhoR [1]. It is possible that during evolution, one of these two-component systems gains new functions to detect phosphate reduction and activate the expression of T4SS effectors.

In conclusion, we found that host cells recognized *E. chaffeensis* Ech_1067 and then upregulated the expression of PIT1, which transports phosphate from ECVs to the cytosol to inhibit bacterial growth. Our findings deepen the understanding of the innate immune responses of host cells to inhibit bacterial intracellular growth and facilitate the development of new therapeutics for HME.

## 4. Materials and Methods

### 4.1. Bacterial and Cell Culture, Plasmids, and Primers

*E. chaffeensis* Arkansas strain was cultured in human acute leukemia THP-1 cell line in RPMI 1640 medium supplemented with 2 mM l-glutamine and 10% fetal bovine serum (FBS) (Every Green, Huzhou, China) at 37 °C in 5% CO_2_ and 95% air, as previously described [34]. *E. coli* strains DH5α and BL21 (DE3) used for DNA cloning and protein expression were cultured in Luria–Bertani broth supplemented with kanamycin (50 μg/mL) as necessary. The plasmids used in this study are listed in Appendix A. The primers used for gene cloning and qRT-PCR are listed in Appendix A.

### 4.2. Isolation of Host Cell-Free E. chaffeensis

*E. chaffeensis* was purified from infected THP-1 cells as described previously with minor modification [26]. Briefly, *E. chaffeensis*-infected THP-1 cells (2 × 10^7^ cells, > 90% infected) were harvested at 600× *g* for 5 min at room temperature. Cells were resuspended in 6 mL cold 1 × SPK buffer (200 mM sucrose, 50 mM potassium phosphate, pH 7.4) with 2 mM l-glutamine and passed through a 23-gauge needle using a syringe on ice for 25 times to disrupt host cell membrane. The undisrupted cells and cell debris were removed by centrifugation at 1000× *g* for 5 min at 4 °C. The supernatant was centrifuged at 10,000× *g* for 10 min at 4 °C. The bacterial pellet was suspended in fresh culture medium.

### 4.3. Immunofluorescence Microscopy

Immunofluorescence microscopy was performed as described previously with minor modification [12]. At 48 h p.i., *E. chaffeensis*-infected THP-1 cells were cytocentrifuged onto glass slides. The cells were fixed with PBS containing 4% paraformaldehyde at room temperature for 10 min, and then blocked and permeabilized with PGS (PBS supplemented with 0.5% bovine serum albumin, 0.1% gelatin, and 0.1% saponin) at room temperature for 1 h. The fixed cells were incubated with rabbit anti-PIT1 antibody (1:100) (Abcam, Cambridge, MA, USA, ab237527) and mouse anti-*E. chaffeensis* FtsZ serum (1:100) diluted in PGS at 4 °C overnight. The cells were washed with PBS for 5 times and then incubated with goat anti-rabbit IgG H&L-conjugated Dylight 488 IgG (1:200) (EarthOx, San Francisco, CA, USA, E032220–01) and goat anti-mouse IgG H&L-conjugated Alexa Fluor 594 IgG (1:500) (Cell Signaling Technology, Boston, MA, USA, 8890S) diluted in PGS at room temperature for 1 h. After being washed with PBS for 5 times, the DNA in host cell nucleus and *E. chaffeensis* was stained with DAPI diluted at 1:500 in PGS at room temperature for 30 min. The cells were washed with PBS for 5 times. The slides were mounted with anti-fluorescent decay sealer, and observed using an OLYMPUS Laser scanning confocal microscopy (OLYMPUS, Tokyo, Japan).

### 4.4. Expression and Purification of Recombinant Proteins

The recombinant proteins were expressed and purified as described previously with minor modification [26]. The DNA fragment encoding *Ech_1067* without signal peptide (1–30 aa) or full-length *tr1* was amplified using specific primers (Appendix A). The amplified fragment was cloned into pET-33b(+) to express recombinant Ech_1067 or Tr1 with N-terminal His-tag (rEch_1067 or rTr1). The ligated plasmids were transformed into *E. coli* DH5α cells, extracted and confirmed by DNA sequencing. *E. coli* BL21 (DE3) cells were transformed with the plasmids and induced to express rEch_1067 or rTr1 with 0.1 mM isopropyl-thio-β-d-galactoside (IPTG, Solarbio, Beijing, China) at 25 °C for 5 h or 1 mM IPTG at 37 °C for 4 h, respectively. All proteins were purified with Ni-affinity chromatography and dialyzed against a stocking buffer (10 mM Tris-HCl, pH 7.5, 1 mM DTT).

### 4.5. Endotoxin Removal and Assay

ToxinEraser Endotoxin Removal Kit (Genscript, Nanjing, China) was used to remove endotoxin from the purified proteins according to the manufacture’ s instructions. The removal efficiency was confirmed by ToxinSensor Chromogenic LAL Endotoxin Assay Kit (GenScript, Nanjing, China).

### 4.6. Transfection and Stimulation of THP-1 Cells

Silence of target genes in THP-1 cells was performed as previously described with minor modification [12,17]. For silence of PIT1, shRNA targeting PIT1 (Appendix A) or a scramble control shRNA was designed and constructed into the lentiviral vector pGMLV-SC5. The plasmids were packaged into lentivirus by Genomeditech (Shanghai, China). The THP-1 cells were transfected according to the manufacture’ s instructions. For silence of p65 or MyD88, THP-1 cells were transfected with siRNA targeting p65, MyD88 (Appendix A) or a scrambled control siRNA using Lipofectamine 3000 according to the manufacturer’s instructions (Invitrogen, Carlsbad, CA, USA). The silence efficiency of PIT1, p65 and MyD88 were determined by Western blotting.

Stimulation of THP-1 cells was performed as previously described with minor modification [24]. 5 × 10^5^ THP-1 cells were incubated with 20 µg/mL polymyxin B sulfate (PB) and rEch_1067 or 20 µg/mL PB and rTr1 at 37 °C for 2 h. THP-1 cells were collected by centrifugation at 500× *g* for 5 min at room temperature, and then the total RNA was extracted.

### 4.7. Quantitative RT-PCR

Total RNA was extracted from each sample and reverse-transcribed to cDNA as described previously with minor modification [12]. The amounts of *E. chaffeensis* 16S rRNA, human *PIT1* and *GAPDH* were determined with qRT-PCR using specific primers (Appendix A) and the ChamQ Universal SYBR qPCR Master Mix (Vazyme, Nanjing, China) on a StepOnePlus Real-Time PCR System (Applied Biosystems, Carlsbad, CA, USA). The expression levels of *PIT1* were normalized against those of *GAPDH*, and the relative numbers of *E. chaffeensis* were determined as the amounts of 16S rRNA normalized against those of *GAPDH*.

### 4.8. Western Blotting

Western blotting was performed as described previously with minor modification [12]. Briefly, to detect the amount of PIT1, the same numbers of THP-1 cells were harvested by centrifugation at 500× *g* for 5 min at room temperature. The pellet was suspended in 1 × SDS sample buffer and boiled for 10 min. The samples were then subjected to 12% SDS-PAGE, transferred to PVDF membrane (Millipore, Co., Cork, Ireland). The protein levels of PIT1 and β-Actin were determined using rabbit polyclonal anti-PIT1 antibody (Abcam, ab237527) and mouse monoclonal anti-β-Actin antibody (Invitrogen, MA5-15739). The membranes were washed with TBST (20 mM Tris-HCl, 150 mM NaCl, 0.05% (V/V) Tween 20) for 4 times, and then incubated with HRP-conjugated goat anti-rabbit IgG H&L secondary antibody (Promega, Madison, WI, USA, W4011) or HRP-conjugated goat anti-mouse IgG H&L secondary antibody (Promega, W4021). The specific bands were detected with an Immobilon Western kit (Millipore, Co., Cork, Ireland) and a molecular imager ChemiDoc XRS+ (BIO-RAD, Hercules, CA, USA).

To detect the silence efficiency of MyD88 and p65, cells were collected after 24 h of transfection. The protein levels of MyD88 and p65 were determined by rabbit polyclonal anti-MyD88 antibody (Cell Signaling Technology, 4283S) and rabbit polyclonal anti-p65 antibody (Cell Signaling Technology, 8242S), respectively.

### 4.9. Statistical Analysis

Statistical analyses were performed using GraphPad Prism 8.0. The statistical significance of a two-group comparison was assessed using Student’s *t*-test (two-tailed). A value of *p* < 0.05 was considered significant.

## Figures and Tables

**Figure 1 ijms-25-07895-f001:**
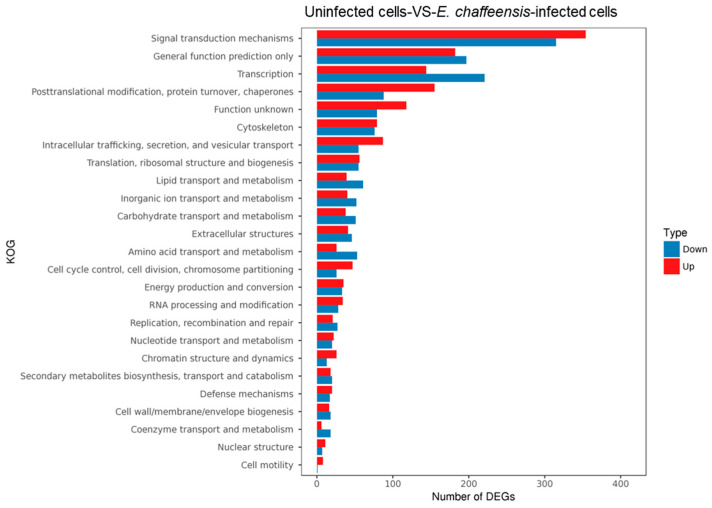
KOG analysis of gene expression between uninfected and *E. chaffeensis*-infected THP-1 cells. The length of bar indicates the number of genes. Red and blue represent significantly upregulated and downregulated genes, respectively (*E. chaffeensis*-infected THP-1 cells vs. uninfected THP-1 cells).

**Figure 2 ijms-25-07895-f002:**
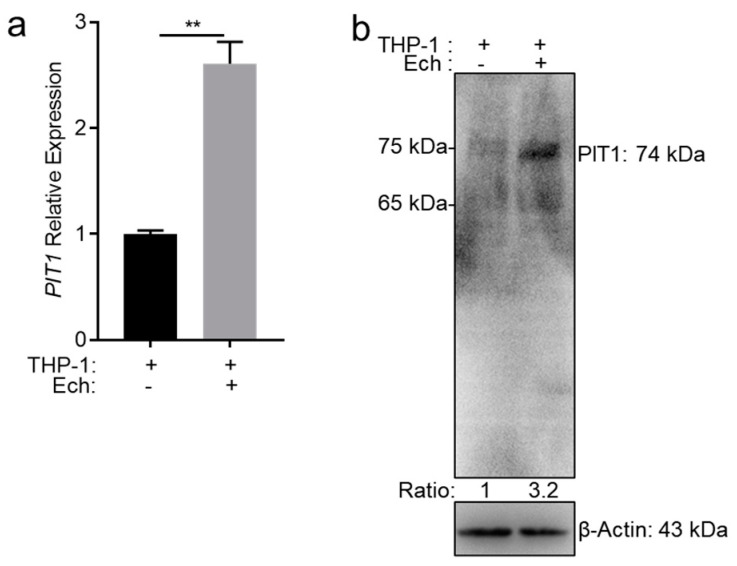
*E. chaffeensis* infection upregulates the PIT1 expression of host cells. (**a**) RNA samples were prepared from uninfected or *E. chaffeensis*-infected THP-1 cells at 48 h p.i. The mRNA levels of *PIT1* were determined with qRT-PCR and normalized against those of human *GAPDH*. Relative values to the *PIT1* amount in control THP-1 cells are shown. Data indicate means  ±  standard deviations (*n* = 3). The significant difference is represented by *p*-values determined with Student’s *t*-test (** *p* < 0.01). (**b**) The protein levels of PIT1 or β-Actin were determined with Western blotting. The number below each panel indicates the relative intensity of each protein band. The protein level of PIT1 in uninfected THP-1 cells is set as 1.

**Figure 3 ijms-25-07895-f003:**
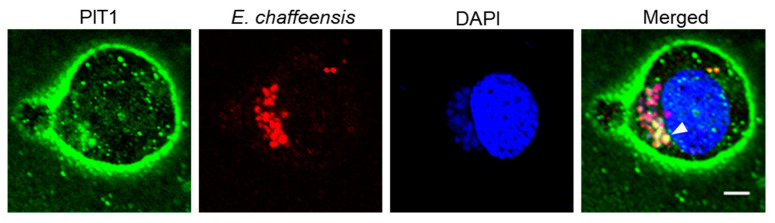
PIT1 is localized on the ECV membrane in infected THP-1 cells. At 48 h p.i., *E. chaffeensis*-infected THP-1 cells were cytocentrifuged onto slides and fixed with 4% PFA. *E. chaffeensis* was labeled with mouse anti-*E. chaffeensis* FtsZ antiserum (red). PIT1 was labeled with rabbit anti-PIT1 antibody (green). The DNA in the host cell nucleus and *E. chaffeensis* was stained with DAPI (blue). The arrow indicates *E. chaffeensis* colocalized with PIT1. Scale bar, 5 µm.

**Figure 4 ijms-25-07895-f004:**
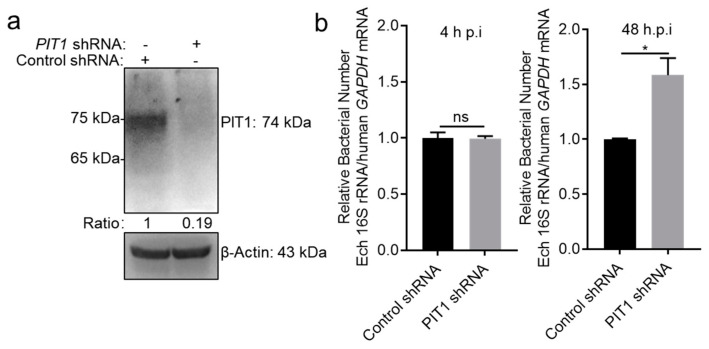
PIT1 inhibits *E. chaffeensis* intracellular growth in host cells. (**a**) The silence effect of PIT1 was determined by Western blotting. The number below each panel indicates the relative intensity of each protein band. The protein level of PIT1 in control cells was set as 1. (**b**) Silence of PIT1 enhances *E. chaffeensis* intracellular growth. THP-1 cells were transfected with shRNA targeting PIT1 or control shRNA and then infected with isolated *E. chaffeensis*. At 4 or 48 h p.i., total RNA was extracted from the cells. The intracellular growth of *E. chaffeensis* was determined with qRT-PCR and normalized against that of human *GAPDH*. Relative values to the *E. chaffeensis* 16S rRNA amount in control THP-1 cells are shown. Data indicate means ± standard deviations (*n* = 3). The significant difference is represented by *p*-values determined with Student’s *t*-test (* *p* < 0.05, ns = not significant).

**Figure 5 ijms-25-07895-f005:**
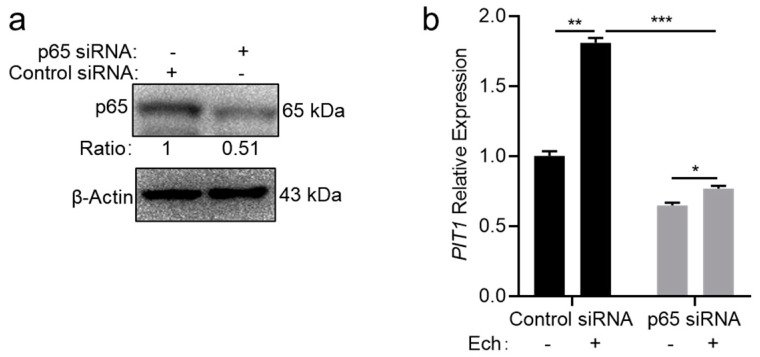
THP-1 cells upregulate the *PIT1* expression in an NF-κB-dependent manner. (**a**) The silence efficiency of p65 was determined by Western blotting. The number below each panel indicates the relative intensity of each protein band. The protein level of p65 in control cells was set as 1. (**b**) Silence of p65 significantly inhibits the upregulation of *PIT1* expression upon *E. chaffeensis* infection. THP-1 cells were transfected with siRNA targeting p65 or control siRNA for 24 h and then infected with isolated *E. chaffeensis*. At 24 h p.i., total RNA samples were extracted from the cells. The mRNA levels of *PIT1* were determined with qRT-PCR and normalized against those of human *GAPDH*. Relative values to the *PIT1* amount in control THP-1 cells are shown. Data indicate means  ±  standard deviations (*n* = 3). The significant differences are represented by *p*-values determined with Student’s *t*-test (*** *p* < 0.001, ** *p* < 0.01, * *p* < 0.05).

**Figure 6 ijms-25-07895-f006:**
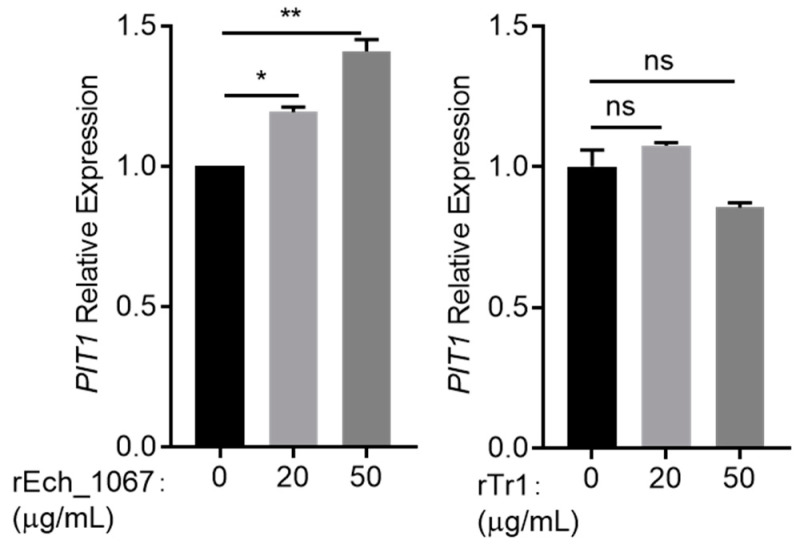
*PIT1* expression is induced by *E. chaffeensis* Ech_1067. THP-1 cells were incubated with PB (20 μg/mL) and rEch_1067 (20, 50 μg/mL) or PB (20 μg/mL) and rTr1 (20, 50 μg/mL) for 2 h, and then total RNA was extracted from the cells. The mRNA levels of *PIT1* were determined with qRT-PCR and normalized against those of human *GAPDH*. Relative values to the *PIT1* amount in THP-1 cells which were treated with PB alone are shown. Data indicate means  ±  standard deviations (*n* = 3). The significant differences are represented by *p*-values determined with Student’s *t*-test (** *p* < 0.01, * *p* < 0.05, ns = not significant).

**Figure 7 ijms-25-07895-f007:**
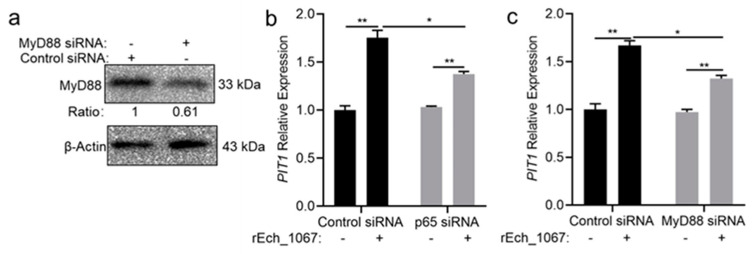
The upregulation of *PIT1* expression induced by rEch_1067 is through the MyD88-NF-κB pathway. (**a**) The silence efficiency of MyD88 was determined by Western blotting. The number below each panel indicates the relative intensity of each protein band. The protein level of MyD88 in control THP-1 cells was set as 1. (**b**,**c**) THP-1 cells were transfected with siRNA targeting p65, MyD88 or control siRNA for 24 h and then treated with PB and rEch-1067 (50 μg/mL) for 2 h. Total RNA was extracted from the cells. The mRNA levels of *PIT1* were determined by qRT-PCR and normalized against those of human *GAPDH*. Relative values to the *PIT1* amount in control THP-1 cells are shown. Data indicate means  ±  standard deviations (*n* = 3). The significant differences are represented by *p*-values determined with Student’s *t*-test (** *p* < 0.01, * *p* < 0.05).

## Data Availability

Transcriptome sequencing raw data have been deposited in the NCBI Short Read Archive (SRA) database (https://www.ncbi.nlm.nih.gov/sra) under the accession number PRJNA1089062, accessed on 31 July 2024. The data presented in the study are included in the article and Appendix A; further inquiries can be directed to the corresponding author.

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
