# Peer review of "Host Cells Upregulate Phosphate Transporter PIT1 to Inhibit Ehrlichia chaffeensis Intracellular Growth"

_ijms, 2024, doi:10.3390/ijms25147895_

Round 1

Reviewer 1 Report

Comments and Suggestions for Authors

Ehrlichia chaffeensis is a Gram-negative obligatory intracellular bacterium that preferentially infects human monocytes or macrophages and causes human monocytic ehrlichiosis (HME). HME is one of the most prevalent, life-threatening emerging zoonoses. The manuscript entitled “Host Cells Upregulate Phosphate Transporter PIT1 to Inhibit Ehrlichia chaffeensis Intracellular Growth” demonstrates that host cells recognize E. chaffeensis penicillin-binding protein Ech_1067 and via the MyD88-NF-κB pathway upregulate the expression of PIT1, which is a phosphate transporter and transports phosphate from E. chaffeensis-containing vesicles to the cytosol, to inhibit bacterial growth. The findings reported by the authors are very important because they deepen the understanding of the innate immune responses of host cells to inhibit bacterial intracellular growth and facilitate the development of new therapeutics for HME. Under the reviewer’s criteria, the experimental procedures and results are well described and the conclusions are justified, nevertheless some adaptations are required to increase its value for the readers. Some minor issues need to be addressed.

Figure 4a: Ratio is missing.

Line 137 and 139: “a” should be “an”.

Line 169: “which treated” should be “which were treated”.

Line 209: references are not needed.

Line 329: “anti-β-actin” should be “anti-β-Actin”.

Comments on the Quality of English Language

The quality of English Language is good.

Author Response

Comments 1: [ Figure 4a: Ratio is missing. ]

Response 1: [ According to the comment, we have made new Figure 4a. ]

Comments 2: [ Line 137 and 139: “a” should be “an”. ]

Response 2: [ According to the comment, we have modified the sentences (line 137 and 139). We have also modified another sentence with the same error (line 132) ]

Comments 3: [ Line 169: “which treated” should be “which were treated”. ]

Response 3: [ According to the comment, we have modified the sentence (line 169). ]

Comments 4: [ Line 209: references are not needed. ]

Response 4: [ According to the comment, we have removed the references (line 209). ]

Comments 5: [ Line 329: “anti-β-actin” should be “anti-β-Actin”. ]

Response 5: [ According to the comment, we have modified the sentence (line 331). ]

Reviewer 2 Report

Comments and Suggestions for Authors

Comments:

In the paper “Host Cells Upregulate Phosphate Transporter PIT1 to Inhibit 2 Ehrlichia chaffeensis Intracellular Growth” by Li et al., the author explained how the host cell recognizes Ehrlichia chaffeensis Ech-1067a penicillin binding protein and show the upregulation of PIT1 expression which is a phosphate transporter and transports phosphate from ECVs to the cytosol to inhibit bacterial growth. The authors have done a good study to prove their points, but their certain comments that need to be addressed before the paper get accepted.

Major Revision:

·         Since EC is mostly dominant in immunocompromised individuals it will be interesting if the author can prove their point in immunocompromised animals.

·         Most of the experiment is invitro it will be interesting of the author can translate their observation in vivo.

·         The author observed upregulation of PIT1 happens via MyD88 pathway. It will be interesting if the author can observe in Myd88-/- animals.

·         Please put full picture of the western blot using clear marker, 74 kDa band is not clear.

Author Response

Comments 1:[ Since EC is mostly dominant in immunocompromised individuals it will be interesting if the author can prove their point in immunocompromised animals.

Most of the experiment is invitro it will be interesting of the author can translate their observation in vivo.

The author observed upregulation of PIT1 happens via MyD88 pathway. It will be interesting if the author can observe in MyD88-/- animals. ]

Response 1: [ The reviewer is right that it will be more informational if we can translate our observation in vivo using immunocompromised or MyD88-/- animals. In this manuscript we focused on discovering the mechanisms using THP-1 cell line. We are planning to perform animal experiments in the near future, which will contribute remarkedly to the study of E. chaffeensis infection. We have added “Since E. chaffeensis is mostly dominant in immunocompromised individuals[5, 6], in vivo experiments using immunocompromised or MyD88-/- animals will provide more comprehensive information for the mechanism demonstrated here.” in the Discussion (line 218-220). ]

Comments 2: [ Please put full picture of the western blot using clear marker, 74 kDa band is not clear. ]

Response 2: [ According to the comment, we have made new Figure 2b and 4a. ]
